# PIM1 phosphorylation of the androgen receptor and 14-3-3 ζ regulates gene transcription in prostate cancer

Sophie E. Ruff[1,2], Nikita Vasilyev [3], Evgeny Nudler[3,4], Susan K. Logan [1,3✉] & Michael J. Garabedian [1,2✉]

PIM1 is a serine/threonine kinase over-expressed in prostate cancer. We have previously shown that PIM1 phosphorylates the androgen receptor (AR), the primary therapeutic target in prostate cancer, at serine 213 (pS213), which alters expression of select AR target genes. Therefore, we sought to investigate the mechanism whereby PIM1 phosphorylation of AR alters its transcriptional activity. We previously identified the AR co-activator, 14-3-3 ζ, as an endogenous PIM1 substrate in LNCaP cells. Here, we show that PIM1 phosphorylation of AR and 14-3-3 ζ coordinates their interaction, and that they extensively occupy the same sites on chromatin in an AR-dependent manner. Their occupancy at a number of genes involved in cell migration and invasion results in a PIM1-dependent increase in the expression of these genes. We also use rapid immunoprecipitation and mass spectrometry of endogenous proteins on chromatin (RIME), to find that select AR co-regulators, such as hnRNPK and TRIM28, interact with both AR and 14-3-3 ζ in PIM1 over-expressing cells. We conclude that PIM1 phosphorylation of AR and 14-3-3 ζ coordinates their interaction, which in turn recruits additional co-regulatory proteins to alter AR transcriptional activity.

[1] Departments of Urology, New York University School of Medicine, New York, NY 10016, USA. [2] Departments of Microbiology, New York University School of Medicine, New York, NY 10016, USA. [3] Departments of Biochemistry and Molecular Pharmacology, New York University School of Medicine, New York, NY 10016, USA. [4] Howard Hughes Medical Institute, New York University School of Medicine, New York, NY 10016, USA. ✉email: susan.logan@nyumc.org; michael.garabedian@nyumc.org

PIM1 is a serine/threonine kinase upregulated in prostate cancer. While PIM1 protein level is very low in benign prostate tissue, PIM1 expression is high in about half of prostate cancer samples[1]. PIM1 has also been linked to a variety of oncogenic processes in both prostate cancer and other cancers, including cell-cycle progression, genomic instability, resistance to chemotherapeutics, and increased tumorigenicity in mouse xenograft models[2–6]. We and others have previously shown that PIM1 phosphorylates AR, the primary therapeutic target in prostate cancer, at S213, affecting the expression of select AR target genes[7,8]. This phosphorylation of AR at S213 is associated with higher-grade tumors, castration-resistant disease, and altered AR transcriptional activity[7,9]. However, the mechanism by which AR phosphorylation at S213 affects AR transcriptional activity is not fully understood.

We have previously identified a set of endogenous PIM1 substrates in LNCaP cells to elucidate the mechanisms by which PIM1 exerts its oncogenic effects[10]. Among the substrates identified was 14-3-3 ζ at serine 64. 14-3-3 ζ has been previously identified as one of the two most amplified genes in metastatic prostate cancer relative to paired local tumors[11]. In addition to gene amplification in over 50% of castration-resistant prostate cancer samples, 14-3-3 ζ protein expression is indicative of poor prognosis in patients[12]. Previous studies also show an interaction between the AR and 14-3-3 ζ, and co-activation of AR by 14-3-3 ζ[13]. However, the mechanism by which 14-3-3 ζ co-activates AR, and the resulting transcriptional outcome, have not been studied.

Here, we sought to characterize how PIM1 phosphorylation of AR and the AR co-activator, 14-3-3 ζ, alters AR transcriptional activity in prostate cancer. 14-3-3 ζ is a regulatory protein that interacts with phosphorylated serine and threonine residues, and its optimal interaction motif closely resembles the PIM1 consensus phosphorylation motif[14]. Noting that PIM1 has been shown to foster interactions between the 14-3-3 proteins and its substrates, we hypothesized that PIM1 phosphorylation of AR mediates its interaction with the co-activator 14-3-3 ζ[15–17]. Finding this to be the case, we used RNA seq and ChIP seq to further characterize the transcriptional changes and chromatin occupancy resulting from PIM1 phosphorylation and 14-3-3 ζ co-regulation of AR. The results indicate a coordinated response in which PIM1 phosphorylation of AR and 14-3-3 ζ alters AR-dependent gene expression.

## Results

**PIM1 phosphorylates the AR and the AR co-regulator, 14-3-3 ζ, and coordinates their interaction.** We and others have previously shown that PIM1 phosphorylates AR at S213[7,8]. To study the effect of this phosphorylation on AR transcriptional activity, we generated a doxycycline-inducible cell line expressing PIM1 in LNCaP cells. We confirmed that in the presence of doxycycline, the cells over-express PIM1. This resulted in an increase in phosphorylation of AR at S213, as detected by the AR pS213 antibody generated in our laboratory as described previously (Fig. 1a)[18].

We recently reported the identification of AR coactivator 14-3-3 ζ as a PIM1 substrate in LNCaP cells[10,13]. To confirm co-activation of AR transcriptional activity by 14-3-3 ζ, we performed a luciferase assay in LNCaP cells using the ARR3-luciferase reporter gene[19]. Upon siRNA knockdown of endogenous 14-3-3 ζ (encoded by the YWHAZ gene), we found a substantial reduction in luciferase activity, consistent with previous reports of 14-3-3 ζ acting as an AR co-activator (Fig. 1b). To validate the phosphorylation of 14-3-3 ζ by PIM1, we used an in vitro phosphorylation assay followed by mass spectrometry (Fig. 1c), as well as a phos-tag gel assay, in which phosphorylated proteins show a shift on Western blot (Supplementary Fig. 1a)[20].

Both assays indicate phosphorylation of 14-3-3 ζ by PIM1, with the predominant phosphorylation site being S64, the site identified in our screen[10]. This validates our previous finding that 14-3-3 ζ S64 is a target of PIM1 phosphorylation.

Because 14-3-3 ζ interacts with phosphorylated serine and threonine residues, we hypothesized that it may co-activate AR by direct interaction with the PIM1 phosphorylation site S213 on AR. The PIM1 phosphorylation site on AR aligns well with the optimal 14-3-3 consensus interaction sequence (Fig. 1d). To test this, we generated the phosphorylation mutant construct AR S213A. In 293 cells, we over-expressed WT AR or AR S213A, in the absence and presence of PIM1 expression. We immunoprecipitated AR and examined whether 14-3-3 ζ associated with AR in a manner dependent on PIM1 and AR S213 (Fig. 1e). We found the strongest co-immunoprecipitation of 14-3-3 ζ with WT AR in the presence of PIM1. This interaction was reduced in AR S213A, supporting the hypothesis that 14-3-3 ζ interacts with AR in a PIM1-dependent manner through AR pS213.

We have also shown that the phosphorylation of 14-3-3 ζ by PIM1 increases endogenous 14-3-3 ζ localization to the nucleus, while PIM1 inhibition reverses this effect (Supplementary Fig. 1b). Similarly, PIM1 has no effect on the phosphorylation mutant 14-3-3 ζ S64A localization to the nucleus (Supplementary Fig. 1c). Because PIM1 is important for 14-3-3 ζ localization to the nucleus, we hypothesized that 14-3-3 ζ interacts with AR in the nucleus to function as a co-regulator. We therefore tested whether the PIM1 phosphorylation site on 14-3-3 ζ, S64, is important for the AR-14-3-3 ζ interaction. Using immunoprecipitation of endogenous AR in LNCaP cells, we found that exogenous WT 14-3-3 ζ co-immunoprecipitates with AR, and this interaction is reduced in cells expressing 14-3-3 ζ S64A (Fig. 1f). Altogether, these results indicate that the PIM1 phosphorylation of both the AR and 14-3-3 ζ are important for their interaction.

**PIM1 over-expression shifts AR and 14-3-3 ζ chromatin occupancy to increase expression of genes involved in extracellular matrix organization, cell adhesion, and cytokine signaling.** We have shown that PIM1 phosphorylation of AR and 14-3-3 ζ coordinates their interaction, and that 14-3-3 ζ co-activates AR in a luciferase assay (Fig. 1e, f, c). To study how PIM1 affects AR and 14-3-3 ζ chromatin occupancy, as well as downstream gene transcription, we performed a series of RNA-seq and ChIP-seq experiments.

To determine how PIM1 over-expression affects gene transcription, we used doxycycline (dox)-inducible PIM1 over-expressing LNCaP cells (Fig. 1a), comparing control cells in the presence of dox (0.5 ng/mL for 24 h) to PIM1 over-expressing cells in the presence of dox to determine how PIM1 over-expression affects gene transcription. We chose to focus on gene expression in 10% FBS containing serum (which contains 0.1 ng/mL testosterone and 0.007 ng/mL DHT)[21]. These levels are enough to maintain growth of LNCaP cells while staying within physiologically relevant levels of androgen. We found that PIM1 over-expression resulted in up-regulation of 498 genes and downregulation of 184 genes ($p < 0.05$ and 1.25 fold change). Metascape pathway analysis indicates that the upregulated genes are involved in extracellular matrix organization, cell adhesion, and cytokine signaling, among other pathways (Fig. 2b)[22]. The genes involved in extracellular matrix organization include proteins such as matrix metalloproteinases, while genes involved in cell adhesion include cytokines, integrins, and members of the collagen family (Fig. 2c). Ultimately, we wanted to identify which of these transcriptional changes could be attributed to PIM1 phosphorylation of AR and 14-3-3 ζ and their co-regulatory interaction. To do this, we performed ChIP-seq on AR, pS213 AR, and 14-3-3 ζ in the

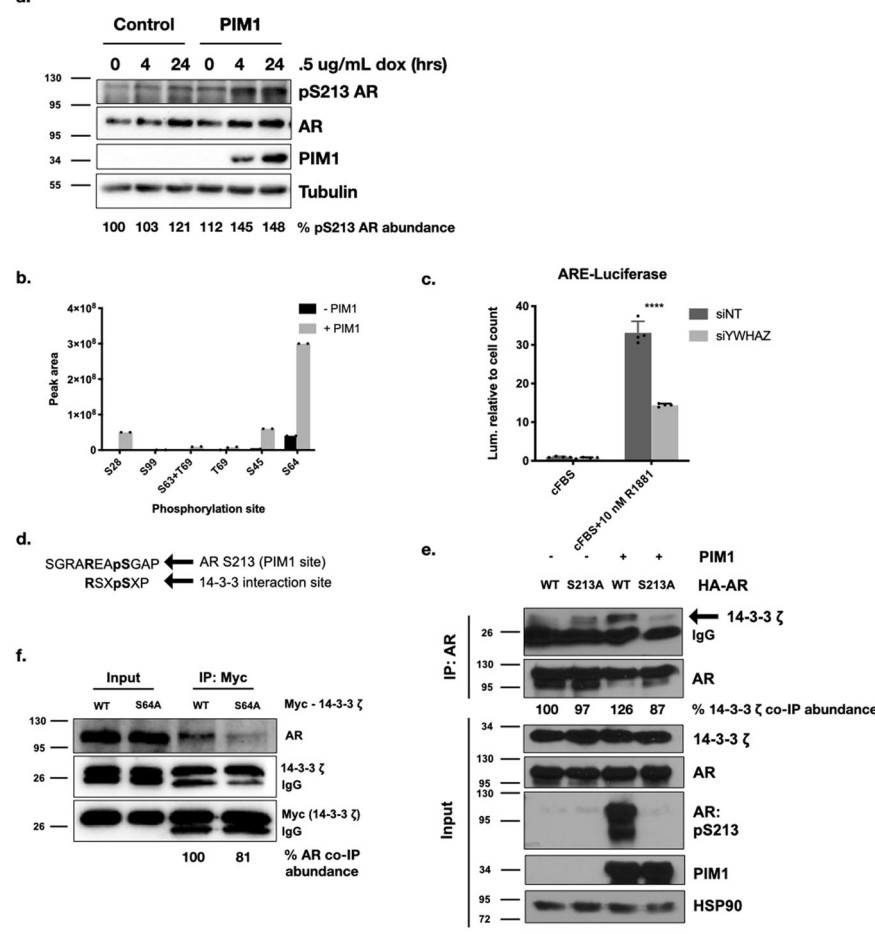

**Fig. 1 PIM1 phosphorylates the AR and 14-3-3 ζ and coordinates their interaction. a** Inducible PIM1 over-expression in LNCaP cells results in an increase in phosphorylation of AR at S213 (including relative abundance of pS213 AR quantified). **b** 14-3-3 ζ functions as an AR co-activator in LNCaP cells. 14-3-3 ζ knockdown in ARR3-luciferase assay leads to reduction in AR transcriptional activity (33.10 +/− 3.008 vs. 14.39 +/− 0.396, p = .000017, n = 4 biological replicates, two sided t test). **c** PIM1 phosphorylates 14-3-3 ζ in vitro, predominately at S64. **d** The PIM1 phosphorylation site on AR, S213, lines up with the 14-3-3 consensus interaction site. **e** 14-3-3 ζ co-immunoprecipitates with AR, particularly in the presence of PIM1 over-expression, and less so with AR phosphorylation mutant S213A (including relative abundance of AR co-IP quantified). **f** AR co-immunoprecipitates with WT 14-3-3 ζ in LNCaP cells, and less so with 14-3-3 ζ phosphorylation mutant S64A (including relative abundance of 14-3-3 ζ co-IP quantified). Error bars refer to standard deviation.

same control and PIM1 over-expressing LNCaP cells. We observed that in PIM1 over-expressing cells, there is a shift in overall AR chromatin occupancy, with about 7% new occupancy sites and a loss of about 25% of occupied sites (Fig. 3a, Supplementary Fig. 2a). When we examined pS213 AR occupancy, there was an increase in the number of sites occupied in the presence of PIM1, which is consistent with the fact that PIM1 increases AR phosphorylation at S213, and gives us confidence in the fidelity of the pS213 AR antibody for ChIP (Fig. 3b, Supplementary Fig. 2b). Additionally, in both the control and PIM1 over-expressing cells, the pS213 AR ChIP almost completely overlaps with the AR ChIP (Fig. 3c, Supplementary Fig 2c). Because the total AR antibody will capture phospho-AR as well as non-phosphorylated AR, the finding that pS213 AR occupies a subset of total AR sites is further validation of the pS213 AR antibody specificity.

When next looked at 14-3-3 ζ chromatin occupancy. We find again that PIM1 over-expression results in a shift in 14-3-3 ζ occupancy, with some sites lost and other sites gained (Fig. 3d, Supplementary Fig 2d). However, the new occupancy sites in the presence of PIM1 have an overall stronger signal (Supplementary Fig. 2d). This is consistent with our hypothesis that PIM1 may drive 14-3-3 ζ into the nucleus and to additional sites on chromatin.

When next examined the sites occupied by 14-3-3 ζ and pS213 AR. We observed in PIM1 over-expressing cells that the majority of pS213 AR sites overlap with 14-3-3 ζ occupancy Fig. 3e, Supplementary Fig. 2e). This finding supports our hypothesis that 14-3-3 ζ co-activates AR by interacting with the PIM1 phosphorylation site on AR, pS213. Performing a Metascape analysis on the top 2000 sites occupied by both 14-3-3 ζ and pS213 AR in PIM1 over-expressing cells, we find that similar gene classes are identified as with the RNA seq analysis: biological adhesion, locomotion, and immune system processes (Fig. 3f)[22]. Although the categories of cellular processes identified in the ChIP-seq analysis appear to be broader than those identified in the RNA-seq analysis, this is likely due to the sheer number of genes showing the AR and 14-3-3 ζ occupancy, compared with the relatively few genes with changes in expression at the RNA level. While there was not complete overlap, we chose to focus on genes in these similar classes to determine whether their expression is dependent on 14-3-3 ζ and AR binding.

**PIM1 mediates upregulation of genes involved in cell migration and invasion.** To validate the gene expression changes in our RNA seq data, we selected a panel of genes involved in extra-cellular matrix organization, cell adhesion, cytokine signaling,

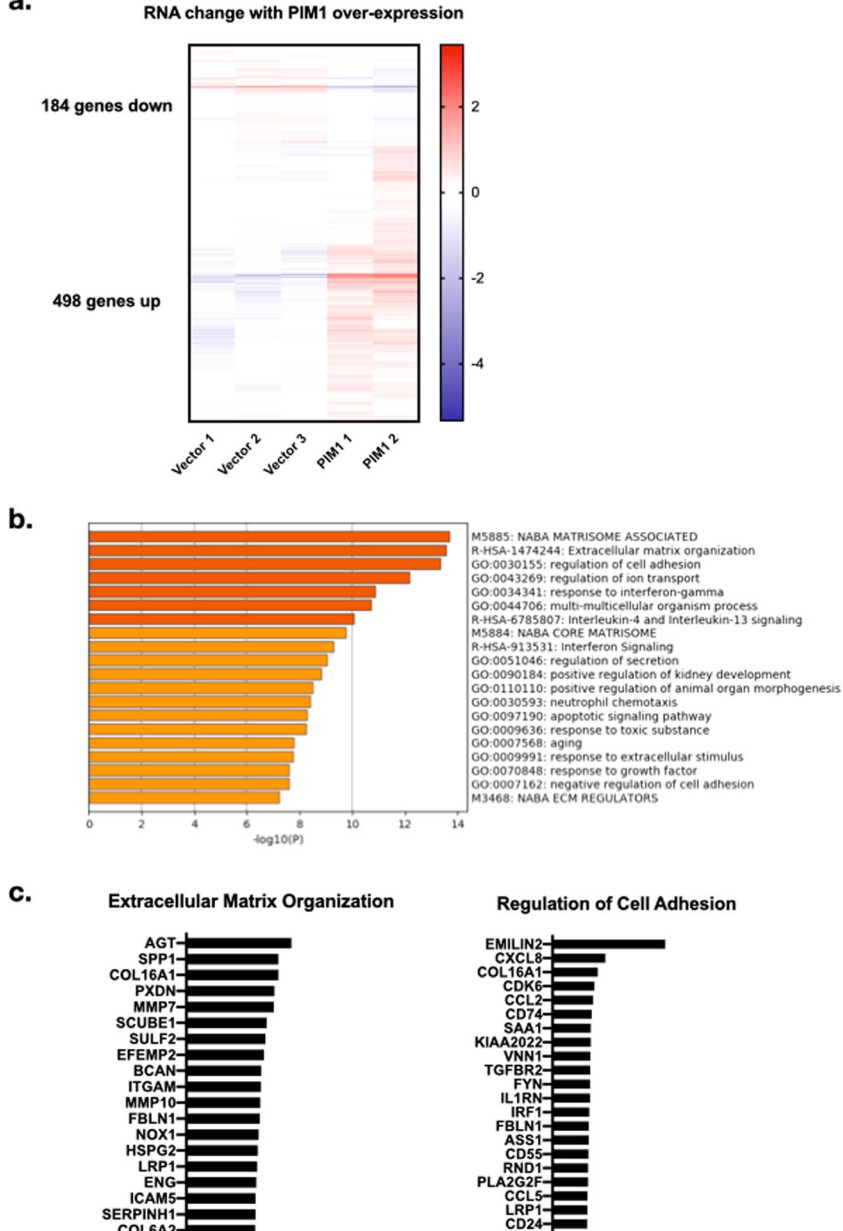

**Fig. 2 PIM1 over-expression in LNCaP cells results in changes in gene expression involved in extracellular matrix organization, cell adhesion, and immune signaling. a** PIM1 over-expression in LNCaP cells results in up-regulation of 498 genes and downregulation of 184 genes (1.25 fold change, *p* value < .05). **b** Metascape analysis of pathways for PIM1 upregulated genes shows that genes are involved in extracellular matrix organization, cell adhesion, and immune system processes. **c** Genes upregulated in extracellular matrix organization and cell adhesion include matrix metalloproteinases, collagen proteins, and immune system factors.

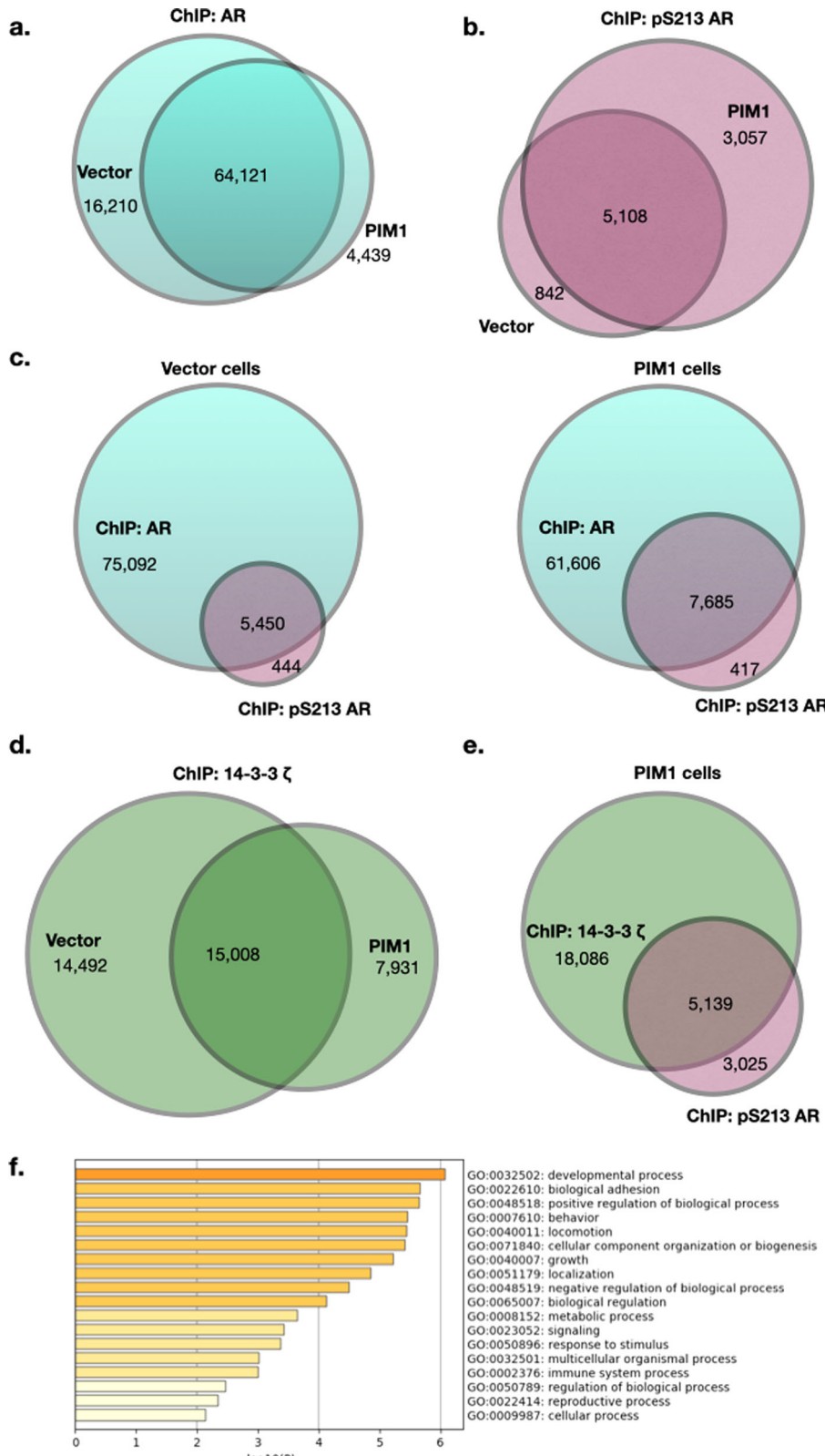

**Fig. 3 14-3-3 ζ and pS213 AR extensively co-occupy chromatin, and PIM1 over-expression shifts their chromatin occupancy. a** PIM1 over-expression results in gain and loss of AR occupied sites on chromatin. **b** PIM1 over-expression results in increased pS213 AR occupancy on chromatin. **c)** pS213 AR occupancy sites almost completely overlap with AR occupancy sites in both control and PIM1 over-expressing cells. **d** PIM1 over-expression shifts 14-3-3 ζ occupancy on chromatin. **e** The majority of pS213 AR sites on chromatin are also co-occupied by 14-3-3 ζ. **f** The top 2000 sites co-occupied by pS213 AR and 14-3-3 ζ in PIM1 over-expressing LNCaP cells are related to biological adhesion, locomotion, and immune system processes.

and cell migration. Altogether, these pathways are key for cell migration and invasion, which is necessary for tumor metastasis. We chose genes which showed chromatin occupancy for either AR or pS213 AR and 14-3-3 ζ, hypothesizing that AR and/or 14-3-3 ζ may play a role in the changes in expression of these genes (Supplementary Fig. 3). These genes include matrix metalloproteinases MMP7 and MMP10. MMP7 (matrilysin) mediates the breakdown of basement membrane proteins, including integrins and E-cadherin[23–25]. Numerous studies have linked MMP7 to prostate cancer progression and metastasis, as well as specifically to promoting bone metastases[26–29]. Although less studied, MMP10 has also been found to be increased at the protein level in metastatic prostate cancer relative to benign tissue[30]. TLL2, also a zinc-dependent metalloproteinase, is a member of the bone morphogenic protein (BMP) family. It has been shown to breakdown laminin-5, and is upregulated in invasive breast cancer[31,32]. While not linked to prostate cancer as of yet, the cytoplasmic dynein protein DYNC1I1 is associated with increased cell migration in gastric cancer, as well as shortened progression-free survival in breast cancer[33,34]. Finally, we included COL4A6, a type IV collagen protein. Collagen proteins are understood as playing both the pro- and anti-tumorigenic roles, depending on the context. They can provide not only a barrier to metastatic cells but also a scaffold which dynamically alters extracellular matrix remodeling and can promote tumor progression, depending on the stage of the tumor[35]. Altogether, we aimed to examine a panel of genes which show AR and 14-3-3 ζ chromatin occupancy, as well as PIM1-dependent expression, to explore the mechanism by which PIM1 may mediate AR-dependent gene expression.

Consistent with the RNA seq data, we found by qPCR the expression of these genes was increased in the PIM1 over-expressing cells (Fig. 4a). To test whether this increase in gene expression was attributable to the catalytic activity of PIM1, we also treated the cells with the PIM1 inhibitor SGI-1776 at 7.5 μM for 24 h. With the exception of DYNC1I1, treatment with the inhibitor resulted in reduced expression of these genes (Fig. 4b). This supports our hypothesis that PIM1 activity results in the up-regulation of these genes.

We next tested whether the PIM1 dependence of these genes' expression identified in LNCaP cells extends to other prostate cancer cell lines. To do this, we treated two unrelated AR-positive prostate cancer cell lines, LAPC4 and 22RV1, with the PIM1 inhibitor SGI-1776. We found that in LAPC4 cells, PIM1 inhibitor resulted in downregulation of 4 of the 5 genes examined, and in 22RV1 cells, it resulted in significant downregulation of two of the five genes examined, as well as a trend towards downregulation in two of the five genes (Fig. 4c). The different response of MMP10 expression to PIM1 inhibitor in these other cell lines could be due to a variety of differences in gene regulation and AR signaling in these cells. These findings suggest a commonality of PIM1 regulation of target gene expression across multiple prostate cancer cells lines.

To determine whether these changes in gene expression alter the phenotype of LNCaP cells, we used transwell assays to measure migration and invasion abilities of control and PIM1 over-expressing LNCaP cells. PIM1 expression increased LNCaP cell migration (Fig. 4d) and invasion through basement membrane proteins (Fig. 4e). This lends functional relevance to the transcriptional changes observed in PIM1 over-expressing LNCaP cells.

**The expression of PIM1-upregulated genes is mediated by 14-3-3 ζ and AR**. Given the occupancy of 14-3-3 ζ and pS213 AR at genes involved in migration and metastasis (Supplementary Fig. 3), we hypothesized that the coordinated action of AR and 14-3-3 ζ via PIM1 phosphorylation increases gene expression. To test this hypothesis, we knocked down 14-3-3 ζ and AR using siRNA in control and PIM1 over-expressing LNCaP cells. 14-3-3 ζ knockdown decreased the expression of each of the genes examined (ns for TLL2) (Fig. 5a). This indicates a role for 14-3-3 ζ in the expression of these genes, which is consistent with our hypothesis.

The knockdown of AR resulted in a decrease in expression for some genes (DYNC1I1 (ns), MMP7, and COL4A6) and, unexpectedly, an increase in expression for other genes [MMP10 and TLL2 (ns)] (Fig. 5b). Although unexpected, we hypothesize that this may be due to AR functioning as an activator at some genes and a repressor at others. This is consistent with previous studies of AR-mediated transcription, which describe AR-mediated gene repression through mechanisms such as recruitment of co-repressors and histone-modifying enzymes[36]. To determine whether AR was mediating the transcription of these genes in a hormone-dependent manner, we also starved the cells of testosterone for 24 h by culturing them in media containing charcoal-stripped serum (cFBS), and treated for 24 h with 10 nM of the synthetic androgen R1881. We hypothesized that genes for which the expression increased with AR knockdown would decrease with R1881 treatment, and vice versa. In fact, we saw this was the case, with DYNC1I1, MMP7 (ns), and COL4A6 showing increased expression with R1881 treatment, and MMP10 and TLL2 showing reduced expression (Fig. 5c). This indicates that AR mediates the transcription of these genes in a hormone-dependent manner. Finally, we treated the cells with the AR-targeting therapeutic enzalutamide. We observed the same response as AR knockdown in most cases, although the effects were not as strong (Fig. 5d). Therefore, each of these genes shows a consistent response to AR knockdown, hormone treatment, and AR inhibition, supporting our hypothesis that AR is involved in PIM1 up-regulation of these genes.

**AR and 14-3-3 ζ occupy chromatin at these PIM1-upregulated genes, and 14-3-3 ζ occupancy is PIM1- and AR-dependent**. Our ChIP seq data shows that 14-3-3 ζ and AR or pS213 AR occupy chromatin at a variety of PIM1 upregulated genes (Supplementary Fig. 3). We examined two of these genes more closely by ChIP-PCR to determine how PIM1 affects 14-3-3 ζ and AR occupancy at these sites. MMP7 and TLL2 both show increased 14-3-3 ζ and pS213 AR occupancy under conditions of PIM1 over-expression in our ChIP-seq data (Fig. 6a).

We validated the increased occupancy of 14-3-3 ζ and AR at the MMP7 and TLL2 genes by ChIP-PCR using primers designed for these regions. We find that both the sites have greater 14-3-3 ζ occupancy in the presence of PIM1 over-expression and that AR co-occupies both sites with 14-3-3 ζ (Fig. 6b). To further characterize the AR and 14-3-3 ζ binding to these sites, we then treated the PIM1 over-expressing cells with the PIM1 inhibitor, SGI-1776, or the AR antagonist, enzalutamide, and examined AR and 14-3-3 ζ occupancy at MMP7. We hypothesized that the PIM1 inhibitor should reduce 14-3-3 ζ binding. Additionally, if 14-3-3 ζ binding is dependent on AR binding, then enzalutamide treatment should reduce both AR and 14-3-3 ζ binding. In fact, we found this to be the case, with reduced 14-3-3 ζ binding to both MMP7 and TLL after treatment either with SGI-1776 (PIM1 inhibitor) or enzalutamide (AR antagonist) (Fig. 6c). This supports our hypothesis that 14-3-3 ζ is recruited to chromatin by AR in a PIM1-dependent manner. As expected, enzalutamide almost completely ablated AR binding to MMP7 (Fig. 6d). However, intriguingly, PIM1 inhibition also reduced AR binding to this site, indicating that not only 14-3-3 ζ binding but also AR binding, may be PIM1-dependent at this site (Fig. 6d).

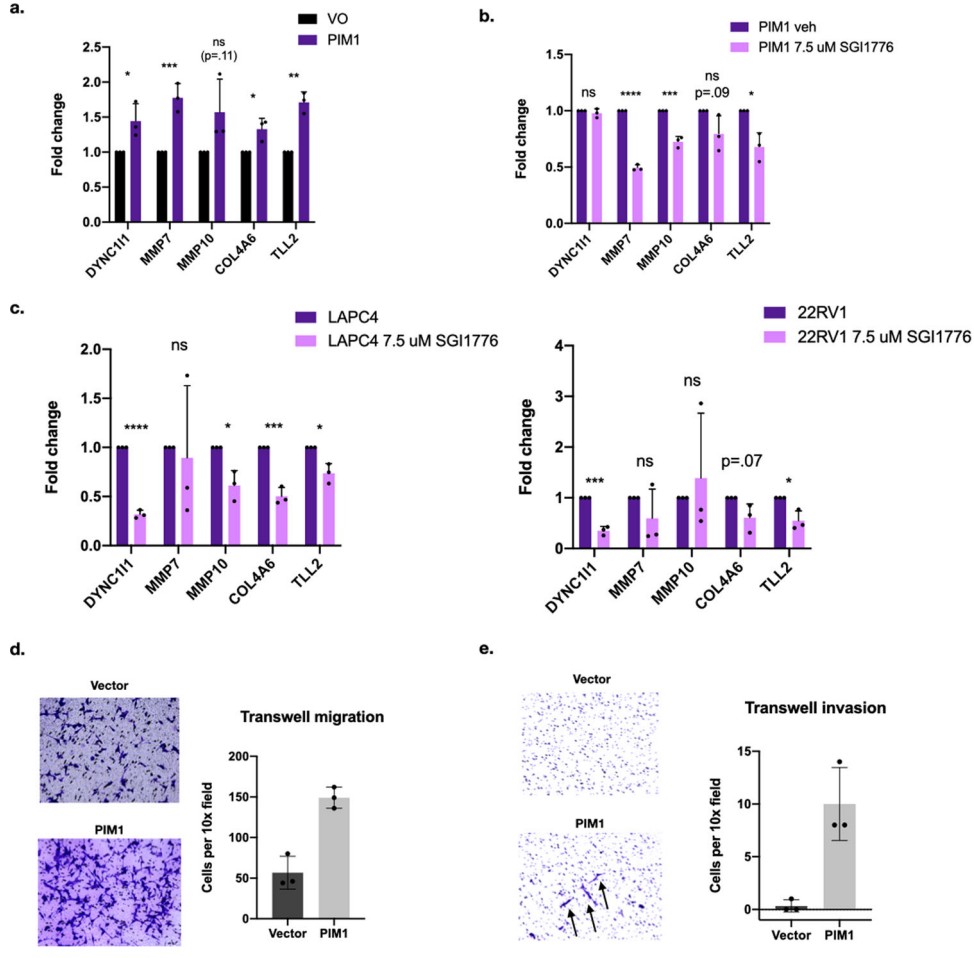

**Fig. 4 PIM1 over-expressing LNCaP cells show increased expression of genes involved in invasion and metastasis. a** Validation of increased expression of genes identified in RNA sequencing in PIM1 over-expressing cells (DYNC1I1: 1.44 +/− 0.25, p = 0.038. MMP7: 1.78 +/− 0.20, p < 0.0027. MMP10: 1.57 +/− 0.47, p = 0.11. COL4A6: 1.33 +/− 0.15, p = 0.021. TLL2: 1.71 +/− 0.15, p = 0.0011. n = 3 biological replicates, two sided T test). **b** While PIM1 over-expression increases expression of these genes, PIM1 inhibition with 7.5 μM SGI1776 for 24 h reduces expression of all genes except for DYNC1I1 (DYNC1I1: 0.98 +/− 0.040, p = 0.39. MMP7: 0.49 +/− 0.026, p < 0.0001. MMP10: 0.73 +/− 0.048, p = 0.00056. COL4A6: 0.79 +/− 0.16, p = 0.091. TLL2: .68 +/− .13, p = 0.011. n = 3 biological replicates, two sided T test). **c** PIM1 inhibition with 7.5 μM SGI1776 for 24 h reduces expression of 4 of the 5 genes in gene panel in LAPC4 cells, and significantly reduces 2 of the 5 genes in 22RV1 cells, with other genes showing a trend towards downregulation (LAPC4: DYNC1I1: 0.32 +/− 0.041, p < 0.0001. MMP7: 0.89 +/− 0.74, p = 0.81. MMP10: 0.61 +/− 0.15, p = .011. COL4A6: 0.50 +/− 0.09, p = 0.00064. TLL2: .74 +/− 0.10, p = .010. n = 3 biological replicates, two sided T test. 22RV1: DYNC1I1: 0.35 +/− 0.085, p = 0.00019. MMP7: 0.59 +/− 0.58, p = 0.29. MMP10: 1.39 +/− 1.28, p = 0.63. COL4A6: 0.61 +/− 0.27, p = 0.066. TLL2: 0.55 +/− 0.20, p = 0.016. n = 3 biological replicates, two sided T test.) **d** PIM1 over-expressing cells show increased migration on transwell migration assay. **e** PIM1 over-expressing cells show increased invasion on transwell invasion assay through extracellular matrix coating. Error bars refer to standard deviation.

**AR and 14-3-3 ζ share interactions on chromatin with transcriptional co-regulators**. To further understand how 14-3-3 ζ may be modulating AR activity, we performed rapid immunoprecipitation mass spectrometry of endogenous proteins for analysis of chromatin complexes (RIME) on AR and 14-3-3 ζ in LNCaP cells in the presence of PIM1[37]. This enabled us to observe what other proteins interact with 14-3-3 ζ and AR on chromatin. For their respective pulldowns, the AR and 14-3-3 ζ were the fifth and fourth most abundant proteins identified on mass spectrometry (Supplementary Data 1). This lends further confidence to the specificity of the antibodies used for the ChIP-seq analysis. Other 14-3-3 family proteins were identified as coimmunoprecipitated with 14-3-3 ζ (Supplementary Data 1, Supplementary Fig. 5a). Although this may be due to some cross reactivity of the antibody, it is likely also attributable to heterodimerization between different 14-3-3 isoforms[38]. In fact, reports of IP-mass spec experiments to identify 14-3-3 ζ binding partners repeatedly pull down abundant quantities of all other 14-3-3

isoforms[39]. In any case, 14-3-3 ζ was the most abundant isoform detected by RIME (Supplementary Fig. 4a).

We examined the top 250 proteins coimmunoprecipitated on chromatin for both the AR and 14-3-3 ζ from 3 replicates of RIME analysis, ranked by average abundance relative to IgG and AR abundance (Supplementary Data 1). We used Metascape analysis to examine the categories of proteins associated with AR (Fig. 7a) and 14-3-3 ζ on chromatin (Fig. 7b)[22]. For AR, we identified many known AR co-regulators, as well as proteins previously identified in RIME experiments for AR. For instance, Stelloo et al. performed RIME analysis on AR, showing validated interactions with known interactors ARID1A (a component of SWI/SNF chromatin remodeling complex), HOXB13 (a homeodomain family transcription factor responsible for androgenic response), HSP90 (a chaperone that promotes the maturation of the AR LBD), FOXA1 (a member of the forkhead class of DNA-binding proteins), and PARP1 (modifies various nuclear proteins by poly(ADP-ribosylation), as well as their novel findings of AR

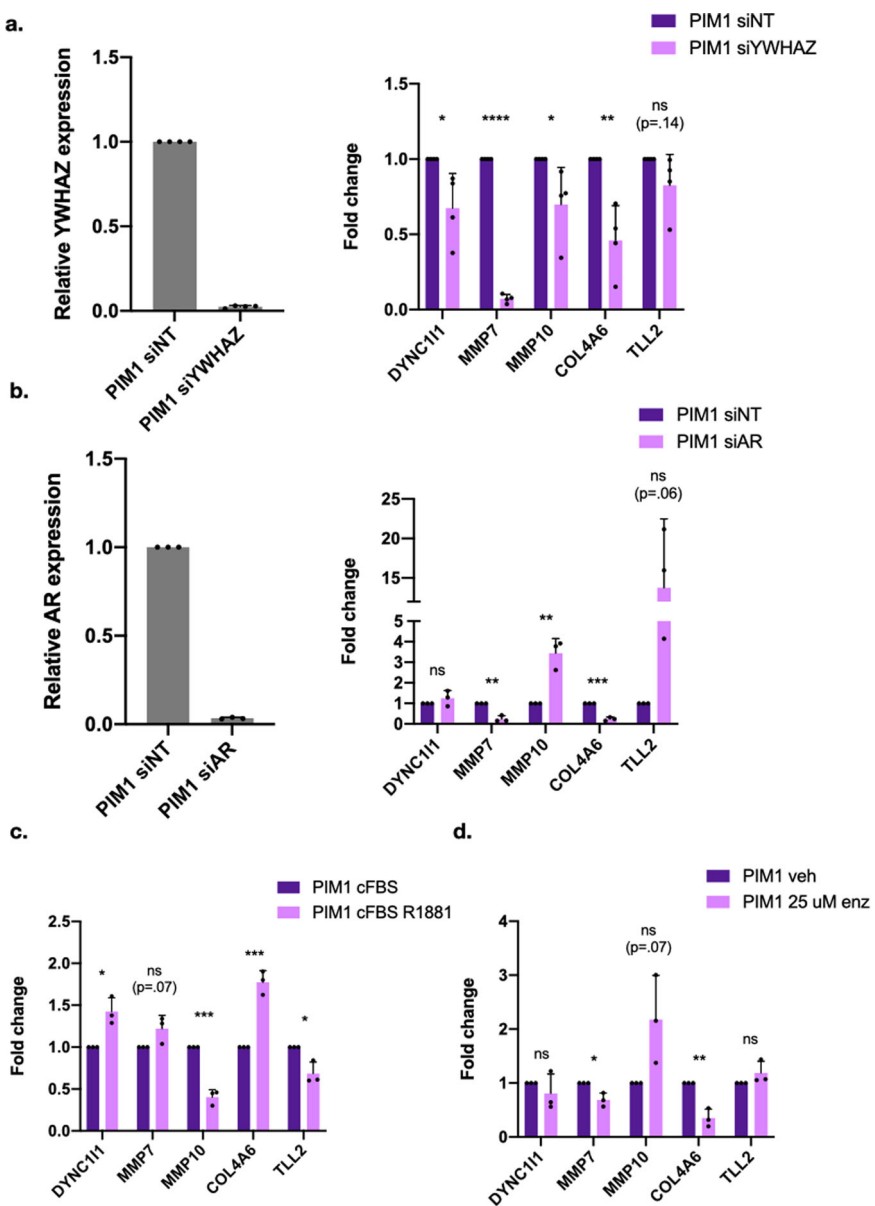

**Fig. 5 Expression of PIM1 upregulated genes are dependent on 14-3-3 ζ and AR expression and activity. a** Knockdown of 14-3-3 ζ results in decrease in expression of PIM1-induced genes involved in cell migration and invasion (DYNC1I1: 0.68 +/− 0.23, p = 0.030. MMP7:0 .072 +/− 0.038, p < 0.0001. MMP10: 0.70 +/− 0.25, p = 0.025. COL4A6: 0.46 +/− 0.23, p = 0.0034. TLL2: 0.83 +/− 0.21, p = 0.14. n = 4 biological replicates, two sided T test). **b** Knockdown of AR results in reduction in expression of some of these genes, but an increase in expression of others, indicating that AR may act as an activator at some genes and a repressor at others (DYNC1I1: 1.25 +/− 0.37, p = 0.30. MMP7: 0.24 +/− 0.16, p = 0.0011. MMP10: 3.44 +/− 0.71, p = 0.0040. COL4A6: 0.24 +/− 0.091, p = 0.00013. TLL2: 13.76 +/− 8.72, p = 0.064. n = 3 biological replicates, two sided T test). **c** Cells in charcoal-stripped media treated with the synthetic androgen R1881 show complementary gene expression changes to cells with AR knockdown (DYNC1I1: 1.43 +/− 0.16, p = .011. MMP7: 1.22 +/− 0.16, p = 0.075. MMP10: 0.40 +/− 0.089, p = 0.00031. COL4A6: 1.78 +/− 0.14, p = 0.00066. TLL2: 0.69 +/− 0.14, p = 0.016. n = 3 biological replicates, two sided T test). **c** 25 uM enzalutamide treatment for 24 h for the most part replicates AR siRNA knockdown (DYNC1I1: 0.81 +/− 0.36, p = 0.41. MMP7: 0.69 +/− 0.13, p = 0.013. MMP10: 2.18 +/− 0.82, p = 0.067. COL4A6: 0.35 +/− 0.17, p = 0.0027. TLL2: 1.19 +/− 0.22, p = 0.21. n = 3 biological replicates, two sided T test). Error bars refer to standard deviation.

interaction with TLE3 (a transcriptional corepressor) and TRIM28 (corepressor with E3 SUMO-protein ligase activity)[40–48]. Among these, we identified AR interacting in at least two of the three replicates with each of these proteins (Supplementary Data 1). This lends further validation to the RIME analysis.

We found 36 proteins that overlapped between the top proteins identified in AR and 14-3-3 ζ pulldowns (Fig. 7c, Supplementary Data 1). A large number of RNA binding proteins were identified, with functions in RNA splicing and RNA stability being the

predominant pathways identified in the overlapping proteins. While AR and 14-3-3 ζ were not in the top 250 interactors for each other, AR was identified as enriched compared with IgG in all three replicates of the 14-3-3 ζ RIME, and 14-3-3 ζ was identified in one of the three replicates of the AR RIME (Supplementary Data 1).

Notably, within the subset of overlapping proteins, several are known AR co-regulators, including TRIM28 and hnRNPK. We hypothesized that to co-regulate AR, 14-3-3 ζ is recruiting or altering interactions between the AR and additional AR co-

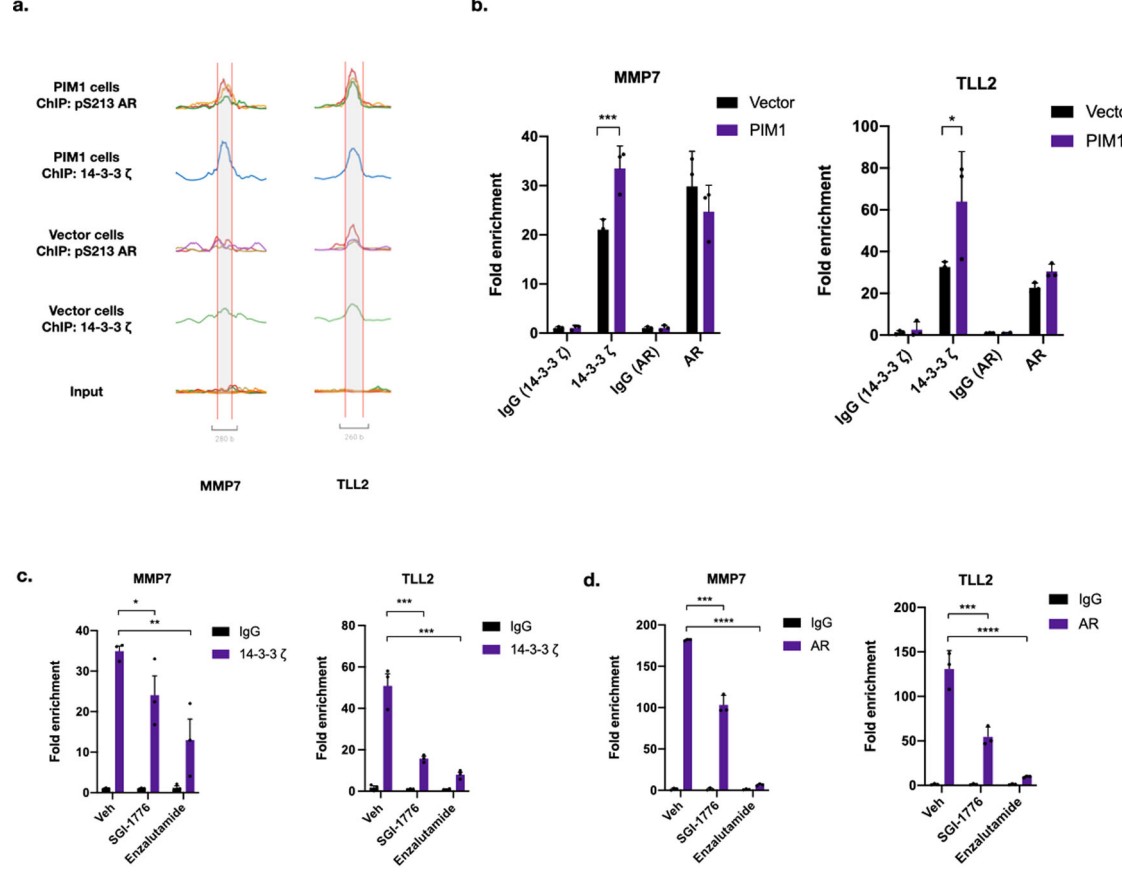

**Fig. 6 AR and 14-3-3 ζ co-occupy chromatin at MMP7 and TLL2 in a PIM1-dependent manner. a** ChIP-seq traces for 14-3-3 ζ and pS213 AR at MMP7 and TLL2 in control and PIM1 over-expressing LNCaP cells. **b** Validation of 14-3-3 ζ and AR binding at MMP7 and TLL2 by ChIP-PCR. 14-3-3 ζ shows increased binding in PIM1 over-expressing cells (MMP7: 33.47 +/− 2.64 vs. 21.07 +/− 1.22, $p = 0.0029$, TLL2: 63.95 +/− 13.81 vs. 32.62 +/− 1.43, $p = .0248$, 3 technical replicates, two sided T test). **c** ChIP-PCR of 14-3-3 ζ binding at MMP7 and TLL2 in PIM1 over-expressing cells after treatment with the PIM1 inhibitor SGI1776 (7.5 uM for 24 h) and enzalutamide (25 uM for 24 h). 14-3-3 ζ binding is reduced in the presence of both treatments for both genes (for MMP7: Veh: 34.95 +/− 1.27 vs. SGI1776 24.09 +/− 4.76, $p = 0.01167$, $n = 3$, two sided T test, Veh: 34.95 +/− 1.27 vs. Enz: 13.02 +/− 5.18, $p = 0.0011$, $n = 3$, two sided T test. For TLL2: Veh: 50.88 +/− 5.78 vs. SGI1776 15.78 +/− 1.02, $p = 0.00057$, $n = 3$, two sided T test, and Veh: 50.88 +/− 5.78 vs. Enz: 8.06 +/− 1.19, $p = 0.00025$, $n = 3$, two sided T test). **d** ChIP-PCR of AR binding at MMP7 and TLL2 in PIM1 over-expressing cells after treatment with the PIM1 inhibitor SGI1776 (7.5 uM for 24 h) and enzalutamide (25 uM for 24 h). AR binding is reduced in the presence of both treatments (for MMP7: Veh: 181.98 +/− 0.53 vs. SGI1776: 103.49 +/− 6.46, $p = 0.000332$, $n = 3$, two sided T test, and Veh: 181.98 +/− 0.53 vs. Enz: 6.78 +/− 0.70, $p = 0.000005$, $n = 3$, two sided T test. For TLL2: Veh: 130.72 +/− 11.96 vs. SGI1776 54.62 +/− 6.20, $p = 0.00012$, $n = 3$, two sided T test, and Veh: 130.72 +/− 11.96 vs. Enz: 9.64 +/− .57, $p < .0001$, $n = 3$, two sided T test). Error bars refer to standard deviation.

regulators. In particular, although hnRNPK has known roles in RNA binding and splicing, studies have also suggested it may have a role in transcriptional co-regulation for AR, as well as other transcription factors such as p53[49,50]. Additionally, TRIM28, identified in AR RIME data from Stelloo et al. was also shown to co-occupy chromatin with AR, and has been identified in other studies to interact with AR and function as a transcriptional co-regulator[40,51,52].

If PIM1 regulates AR-14-3-3 ζ interactions, which in turn recruits co-factors such as hnRNPK and TRIM28 to control gene expression, then we hypothesized that we may see clinical correlation in amplification of YWHAZ (gene that encodes 14-3-3 ζ), PIM1, and one or more of these co-factors. Using available studies of metastatic castration-resistant prostate cancer from cBioPortal, we found that PIM1 and YWHAZ amplification significantly co-occur ($p < .001$), and they both also significantly co-occur with hnRNPK amplification (for PIM1 and hnRNPK: $p < .001$, for YWHAZ and hnRNPK: $p = .015$) (Supplementary Fig. 4b–c)[53–57].

We hypothesized that if TRIM28 and hnRNPK are in complex with AR and 14-3-3 ζ on chromatin, we may see alterations in

PIM1-dependent gene expression with perturbation of hnRNPK and/or TRIM28. In fact, we find that the knockdown of both factors results in changes in PIM1-dependent gene expression (Fig. 8a). With hnRNPK knockdown, DYNC1I1 expression increases (ns), while MMP7, COL4A6, and TLL2 expression decrease. With TRIM28 knockdown, DYNC1I1 (ns), MMP7, and MMP10 expression increase. We hypothesized that TRIM28 and hnRNPK may function as AR co-regulators at these sites. If this were the case, we would expect occupancy of TRIM28 and hnRNPK in the same regions of these genes as AR and 14-3-3 ζ. To test this, we performed ChIP on hnRNPK and TRIM28 using previously validated ChIP antibodies[52,58]. Because MMP7 appears to be regulated by both the hnRNPK and TRIM28 expression, decreasing with hnRNPK knockdown and increasing with TRIM28 knockdown, we tested whether these co-factors occupy the AR and 14-3-3 ζ binding site on MMP7. We found that in PIM1 over-expressing LNCaP cells, both factors show occupancy at this site relative to their respective IgG controls (Fig. 8b). Due to the high occupancy of TRIM28 at this site, and the increased MMP7 expression with TRIM28 knockdown, we performed co-immunoprecipitation of TRIM28 with AR in control and PIM1 over-expressing cells to

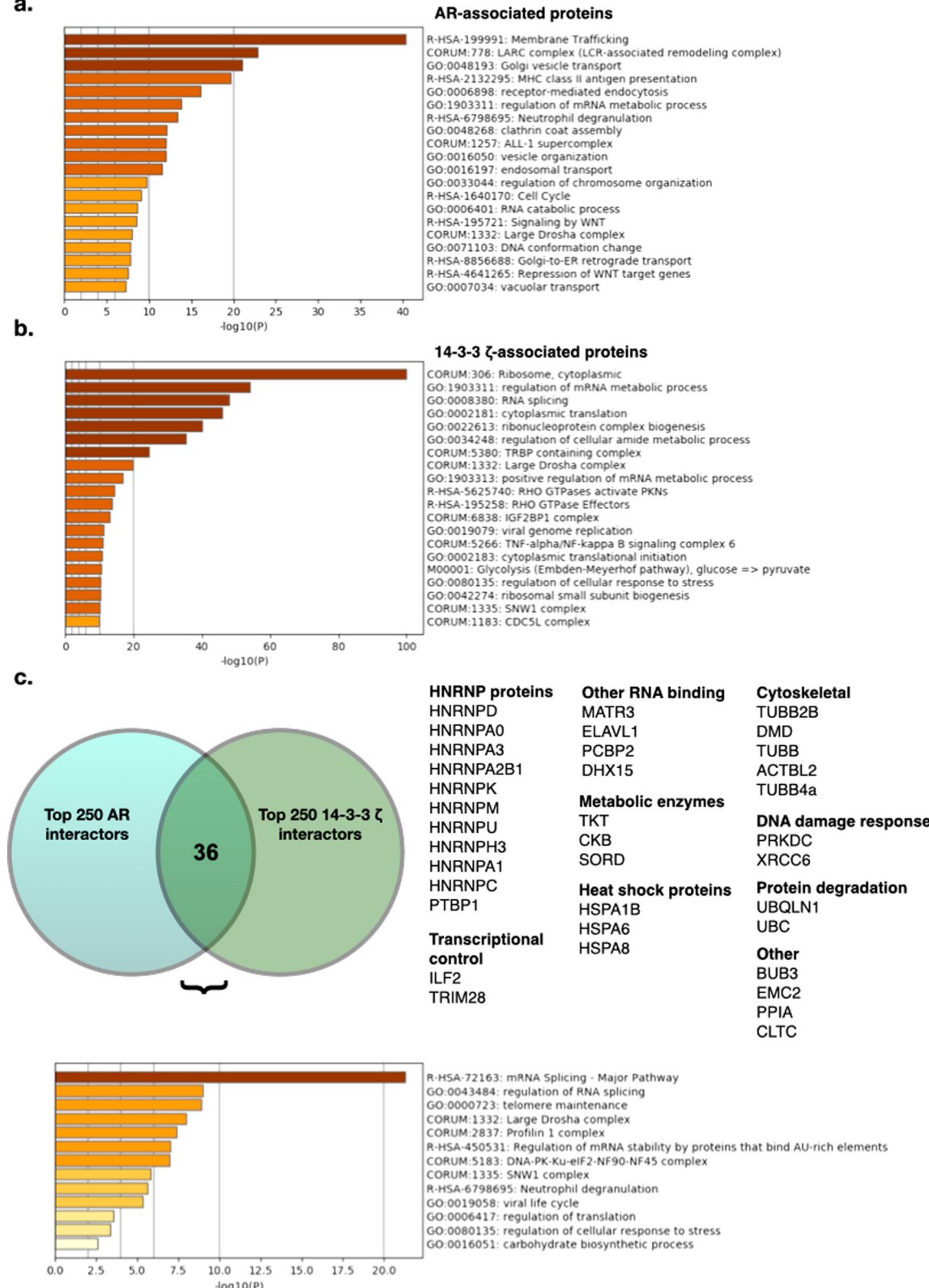

**Fig. 7 AR and 14-3-3 ζ share interactions on chromatin with a variety of proteins, including AR co-regulators. a** AR RIME in LNCaP cells indicates interactions with proteins involved in a variety of processes, including mRNA processing and transcriptional pathways including the LARC complex and WNT signaling. **b** 14-3-3 ζ interacts on chromatin with proteins mainly involved in ribosomal and RNA processing, as well as a variety of other complexes. **c** The overlap between 14-3-3 ζ and AR interactions in PIM1 over-expressing LNCaP cells includes 36 proteins, including HNRNP proteins, other RNA binding proteins, and known AR co-regulators such as hnRNPK and TRIM28. Proteins involved in mRNA splicing are the main pathways identified.

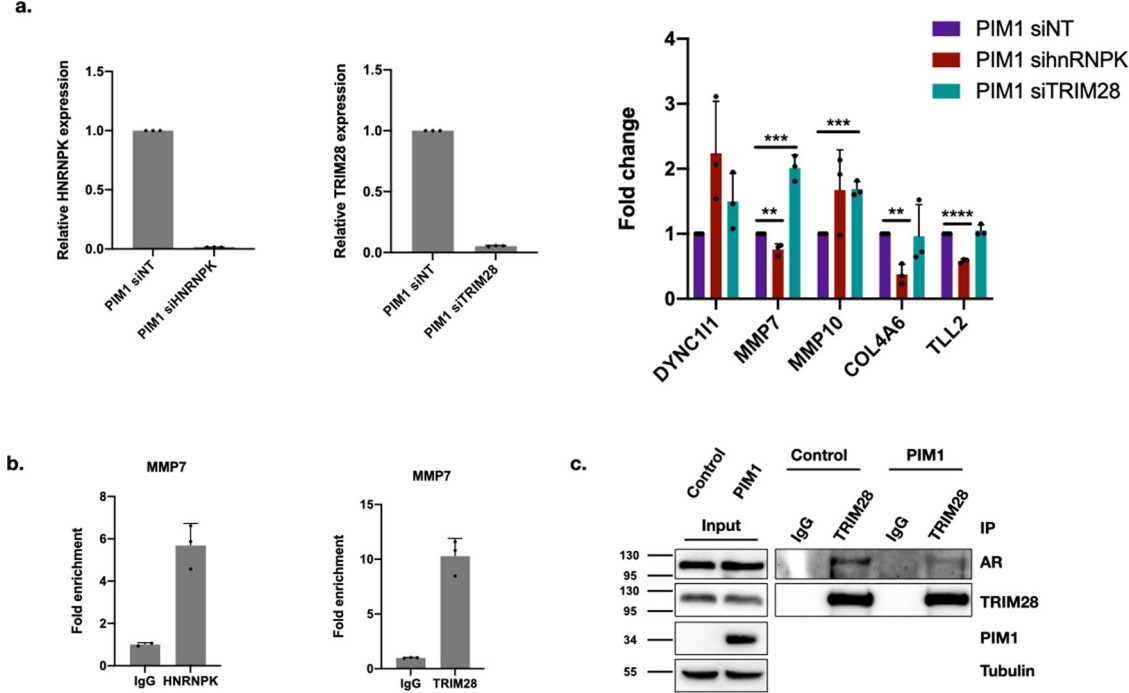

**Fig. 8 AR and 14-3-3 ζ interacting proteins hnRNPK and TRIM28 are involved in PIM1-dependent gene expression. a** Knockdown of hnRNPK and TRIM28 result in alterations of gene expression for PIM1-dependent gene panel (for hnRNPK knockdown: DYNC1I1: 2.24 +/− 0.80, p = .055, MMP7: 0.76 +/− 0.088, p = 0.0087, MMP10: 1.67 +/− 0.62, p = .13, COL4A6: 0.37 +/− 0.16, p = .0023, TLL2: 0.58 +/− 0.032, p < .0001. For TRIM28 knockdown: DYNC1I1: 1.50 +/− 0.43, p = .12, MMP7: 2.01 +/− 0.20, p = 0.00086, MMP10: 1.69 +/− 0.11, p = .00049, COL4A6: 0.96 +/− 0.49, p = 0.90, TLL2: 1.05 +/− .087, p = 36). **b** ChIP of hnRNPK and TRIM28 at MMP7 reveals modest occupancy of hnRNPK at MMP7 AR occupancy site, and substantial TRIM28 occupancy. **c** Co-immunoprecipitation of AR with TRIM28 reveals a reduction in AR interaction with TRIM28 in PIM1 over-expressing cells. Error bars refer to standard deviation.

determine whether PIM1 alters this interaction. We were able to validate the AR-TRIM28 interaction (Fig. 8c), and found that PIM1 over-expression results in a slight reduction of the interaction. This suggests that TRIM28 may act as a corepressor at MMP7, and PIM1 over-expression reduces its interaction with AR, de-repressing MMP7 expression.

## Discussion
Through our studies, we have further characterized the mechanism by which PIM1 phosphorylation of AR alters AR transcriptional activity in prostate cancer. Although previously shown to alter the expression of some AR target genes in cell culture and in patient samples, the mechanism by which phosphorylation of AR S213 may do this was previously unknown[7,9]. Through our previous work, we identified the AR co-activator 14-3-3 ζ S64 as an additional target of PIM1 phosphorylation in prostate cancer[10]. Now, using co-immunoprecipitation, we have shown an interaction between 14-3-3 ζ and AR, which is dependent on these phosphorylation sites. This led us to hypothesize that PIM1 coordination of the interaction between the AR and 14-3-3 ζ may provide a mechanism for how AR phosphorylation at S213 alters its chromatin occupancy and transcriptional activity.

We used ChIP-seq to evaluate the phospho-cistrome of AR pS213 using a phosphorylation site-specific antibody. This, along with RNA seq data and ChIP-seq studies of total AR and 14-3-3 ζ, allowed us to show that the majority of AR pS213 are co-occupied by 14-3-3 ζ. This co-occupancy correlates changes in gene expression, which are dependent on PIM1 activity, 14-3-3 ζ expression, and AR activity. One example of these gene expression changes includes a category of genes involved in cell migration and invasion, including matrix metalloproteinases,

such as MMP7, MMP10, and TLL2. As a result, the PIM1 over-expressing LNCaP cells show increased cell migration and invasion on transwell assays. However, these are only a subset of the genes showing expression changes in PIM1 over-expressing cells. In addition, many genes involved in immune signaling show upregulation (Fig. 2b). PIM1 over-expression has been shown to be associated with an inflammatory phenotype in both breast and male reproductive tumors, including prostate cancer[59,60]. Further studies of our ChIP-seq data and the effect of AR and 14-3-3 ζ on PIM1-dependent inflammatory genes could shed light on the mechanism by which PIM1 causes these gene expression changes.

One intriguing finding is that many of the AR and 14-3-3 ζ binding sites we have identified do not occur at promoters, yet gene expression appears to be AR and 14-3-3 ζ dependent. However, this is in fact consistent with the fact previous ChIP-seq studies of AR indicate that 90% of AR binding sites lie in intronic and other nonpromoter enhancer regions[61–63]. For this reason, AR target genes are determined based on AR binding in proximity to genes, which show transcriptional changes. Our experiments indicate that the genes we examine show not only AR and/or pS213 AR binding in proximity but also gene expression changes related to PIM1, 14-3-3 ζ, and AR expression and inhibition.

Based on our findings, we model a scenario in which PIM1 phosphorylation of AR and 14-3-3 ζ coordinates their interactions, resulting in a shift in AR and 14-3-3 ζ chromatin occupancy, AR-dependent co-occupancy of chromatin by AR and 14-3-3 ζ, recruitment of additional co-regulators, and alterations in gene transcription (Fig. 9). Based on our finding that PIM1 inhibition also reduces AR occupancy (Fig. 6c), we hypothesize that phosphorylation or 14-3-3 ζ co-regulation also may result in the stabilization or increased binding of AR at certain sites.

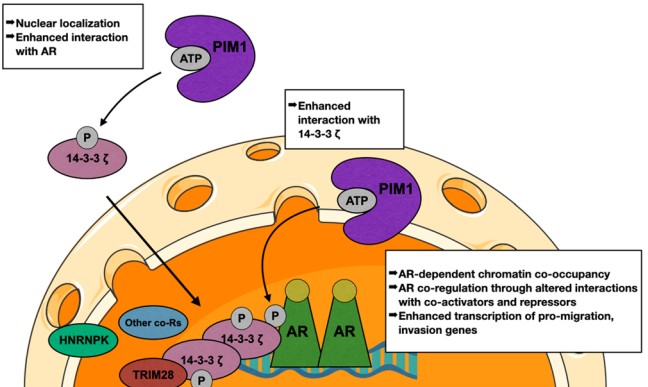

**Fig. 9 Model for PIM1-mediated AR co-regulation by 14-3-3 ζ.** PIM1 phosphorylation of the AR and 14-3-3 ζ enhances their interaction and shifts their occupancy on chromatin, resulting in 14-3-3 ζ co-regulation of AR, likely by recruiting other AR co-regulators such as hnRNPK and TRIM28. Nucleus image used from smart.servier.com.

14-3-3 ζ typically exists predominantly as a homo- or heterodimer with other 14-3-3 isoforms[39,64]. Due to this, one model for 14-3-3 ζ regulation of its binding partners is that one subunit binds one protein and the other subunit binds another, promoting an interaction between its binding partners. In this way, 14-3-3 ζ dimers may interact with both the AR on one subunit and AR co-regulators on another to alter AR transcriptional activity. Our RIME data indicates that both AR and 14-3-3 ζ interact with the AR co-regulators hnRNPK and TRIM28, and we hypothesize that 14-3-3 ζ may mediate this interaction in a PIM1 phosphorylation-dependent manner. Similarly, due to its ability to homodimerize, 14-3-3 ζ interactions with its binding partners can promote the binding partner's multimerization[65]. This could explain the reduction in AR chromatin binding after treatment with PIM1 inhibitor: reduction in phosphorylation leads to a breakdown of the AR-14-3-3 ζ interaction, leading to loss of AR dimerization and a reduction in local AR concentration near this chromatin binding site.

Altogether, our findings shed light on the mechanism by which PIM1 phosphorylation of AR at S213 may alter its transcriptional activity through altering its interaction with 14-3-3 ζ, and, consequently, additional co-regulatory proteins. This results in transcriptional changes which alter extracellular matrix remodeling genes, cell adhesion genes, and cytokine signaling. Our findings provide an example for how phosphorylation of transcription factors can alter co-regulatory complexes to modulate their transcriptional activity.

## Methods

**Cell culture and reagents**. LNCaP, 293, and 22RV1 cells were obtained from ATCC. LAPC4 cells were a gift from Dr. R. Reiter (University of California, Los Angeles). 293 cells were cultured in DMEM, LAPC4 cells were cultured in IMDM, and 22RV1 and LNCaP cells were cultured in RPMI (containing phenol red and L-glutamine), all with 10% FBS and 1% penicillin/streptomycin. All cell lines were regularly assessed for mycoplasma contamination. For chemical treatments, the following were used: doxycycline (Sigma), SGI-1776 (Selleck Chemicals), and Enzalutamide (MedKoo).

**Plasmids, cloning primers, and antibodies**. Constructs used were as follows: pCW57-MCS1-P2A-MCS2 (Neo) PIM1, backbone a gift from Adam Karpf, Addgene plasmid #89180, insert cloned from plasmid previously described in Ha et al[7] using HiFi kit (NEB) (forward primer: GCCTGGAGAATTGGCTAGCGAT GGATTACAAGGATGACGAC, reverse primer: AAGGCGCAACCCCAACCCCG CTATTTGCTGGGCC), pcDNA3-HA-AR (WT and S213A, previously described in Ha et al[14], and pcDNA3.1-Myc-14-3-3 ζ, a gift from Huda Zoghbi (Addgene Plasmid #48798). For stable 14-3-3 ζ lentiviral infection in LNCaP cells, 14-3-3 ζ was

cloned into pCW57-MCS1-P2A-MCS2 (Neo) using HiFi kit (NEB) (forward primer: GCCTGGAGAATTGGCTAGCGATGGATAAAAATGAGCTGGT, reverse primer: AAGGCGCAACCCCAACCCCGTTACAGATCCTCTTCTGAGA). S64A mutation was introduced to 14-3-3 ζ plasmid using QuikChange Lightning Multi Site-Directed Mutagenesis Kit (Agilent), with the single primer CTTCCGTCTTTTGTTCAATA GCTGAGACGACCCTCCAAGATG. All plasmids were sequenced to confirm the mutations and to ensure no additional changes in the sequence were introduced.

Commercial antibodies used were as follows: PIM1 (sc-13513; Santa Cruz Biotechnology), 14-3-3 ζ (Western blot and IP for Western blots: sc- 293415; Santa Cruz Biotechnology, ChIP seq and RIME: ab51129; Abcam), AR (Western blot and IP for Western blots: sc-7305; Santa Cruz Biotechnology, ChIP-seq and RIME: blend of sc-7305; Santa Cruz Biotechnology and #5153; Cell Signaling), hnRNPK (ab39975, Abcam), TRIM28 (ab10483, Abcam), HSP90 (610418, BD Biosciences), Tubulin (MMS-489P, Covance), Myc-tag (#2276, Cell Signaling), SP1 (PIPA529165, Fisher Scientific), and BRG1 (ab4081, Abcam). Immunoprecipitations for Western blot were carried out using Protein A/G PLUS-agarose beads (SCBT).

**Transfection and transduction**. Transient plasmid transfections were done using Lipofectamine 2000 (Invitrogen) according to the manufacturer's protocol. siRNA transfections (Dharmacon, OnTarget SMARTpools) were done using Lipofectamine RNAi Max (Invitrogen) according to the manufacturer's protocol. Viral transductions were carried out in 293 T cells using 5 µg lentiviral construct, 2 µg pMD2.G (a gift from Dider Trono, Addgene plasmid #12259), and 3 µg psPAX2 (a gift from Didier Trono, Addgene plasmid #12269) using Lipofectamine reagent (Invitrogen). Target cells were treated with viral supernatant and polybrene (8 µg/mL, H9268, Sigma) overnight, and media replaced. Pools selected using Geneticin (ThermoFisher, 10-131-035, 50 µg/mL).

**Luciferase assay**. Luciferase assays were conducted using stably transfected ARR3-luciferase LNCaP cells. Cells transfected with siRNA were seeded into 384-well plates at 5000 cells/well, followed by 24 h treatment with R1881. Both were measured using the ONE-Glo detection kit (Promega) and data are standardized to CellTiter-Fluor (Promega) viability readings to account for potential variations in well to well cell number. Four biological replicates were used for the experiment.

**In vitro phosphorylation assay**. Three microgram of recombinant 14-3-3 ζ (Abcam, ab87361) were combined with 250 ng of recombinant PIM1 kinase (Abcam, ab60835) in a buffer containing 20 mM MOPS pH 7.2, 5 mM EDTA, 25 mM glycerol-2-phosphate, 1 mM DTT, 500 µM ATP, and 75 mM MgCl2 for 15 min at 30 °C. The experiment was performed in duplicate. Proteins (~3 µg) from phosphorylation reactions were precipitated using 4 vol. of acetone at −20 °C for 1 h. Acetone pellets, collected by 10-min centrifugation at 16,000 g, were rinsed with 80% acetone and dried on air. Dried pellets were dissolved in 5 µl of denaturing buffer: 50 mM NH₄HCO₃, 5 mM DTT, 8 M urea. For cysteine alkylation, samples were mixed with 0.5 µl 0.15 M iodoacetamide followed by 30-min incubation at room temperature in dark. Alkylation reactions were then mixed with 50 µl of the solution containing: 50 mM NH₄HCO₃, 5 mM DTT, and 10 ng/µl trypsin/Lys-C mix (Promega), followed by overnight incubation to complete proteins digestion. Digestion reactions were mixed with 5 µl 20% heptafluorobutyric acid and peptides were desalted using C18 spin tips according to manufacturer protocol (Thermo Scientific). Desalted peptides were dried under vacuum and reconstituted in 12 µl of 0.1% formic acid before LC-MS analysis on Orbitrap Fusion Lumos mass spectrometer. Peptides concentration was measured spectrophotometrically at 205 nm using NanoDrop One (Thermo Scientific).

**LC-MS analysis of peptides**. Peptides were analyzed by LC-MS on Orbitrap Fusion Lumos mass spectrometer coupled with Dionex Ultimate 3000 UHPLC. During each run, 0.5-2 µg of peptides from individual samples were injected and resolved on 50-cm long EASY-Spray column (Thermo Scientific) by 90-min long linear gradient of 4–40% acetonitrile in 0.1% formic acids at flowrate 0.25 µl/min. Method for the data-dependent acquisition was based on published protocol with exception of each cycle was set to last for 2 s instead of 3 s[66].

Peptides identification and label-free quantitation were done in Proteome Discoverer 2.1 or MaxQuant 1.6. Search engines, Sequest and Andromeda respectively, were supplied with a protein database consisting of human or mouse proteome downloaded from UniProt (www.uniprot.org) combined with a set of known protein contaminants (supplied with MaxQuant). Parameters were set to search for peptides of at least five amino acids long, containing at most 2 missed trypsin cleavages, and, depending on the experiment, having dynamic modifications, including carbamidomethylation of cysteine, oxidation of methionine, phosphorylation of serine, threonine, or tyrosine, acetylation of lysine and protein N-terminus, mono- and dimethylation of lysine and arginine. In MaxQuant, the intensity determination option was set to "Total sum" and "Match between runs" option was enabled. All the other parameters were left at their defaults. For analysis, each sample was normalized to protein A/G abundance. IgG-bound peptides were subtracted from antibody-bound peptides, and then samples were normalized to the amount of AR or 14-3-3 ζ pulled down in each replicate.

**Phos-tag gel assay**. Phos-tag gels were prepared with gel containing 40 µM of Phos-tag Acrylamide (Wako) and 80 µM MnCl$_2$. Samples were prepared in EDTA-free RIPA and treated with calf intestinal phosphatase (ThermoFisher) at 1 unit per microgram of protein for 30 min at 37 °C for 30 min before adding loading buffer.

**RNA-sequencing**. RNA was extracted using the RNeasy kit (Qiagen), and libraries were prepared for sequencing using the NEBNext Ultra II RNA Library Prep Kit for Illumina with polyA selection. Concentrations were measured using Qubit and checked for quality using Tapestation, then sequenced on Illumina NovaSeq 6000 SP100 flowcell. Data were analyzed by ROSALIND® (https://rosalind.onramp.bio/), with a HyperScale architecture developed by OnRamp BioInformatics, Inc. (San Diego, CA). Reads were trimmed using cutadapt[67]. Quality scores were assessed using FastQC[68]. Reads were aligned to the Homo sapiens genome build hg19 using STAR3[69]. Individual sample reads were quantified using HTseq and normalized via Relative Log Expression (RLE) using DESeq2 R library[70,71]. Read Distribution percentages, violin plots, identity heatmaps, and sample MDS plots were generated as part of the QC step using RSeQC[72]. DEseq2 was also used to calculate fold changes and $p$ values and perform optional covariate correction. Clustering of genes for the final heatmap of differentially expressed genes was done using the PAM (Partitioning Around Medoids) method using the fpc R library[73]. Hyper-geometric distribution was used to analyze the enrichment of pathways, gene ontology, domain structure, and other ontologies. The topGO R library was used to determine local similarities and dependencies between the GO terms in order to perform Elim pruning correction[74]. Several database sources were referenced for enrichment analysis, including Interpro, NCBI, MSigDB, REACTOME, and WikiPathways[75–80]. Enrichment was calculated relative to a set of background genes relevant for the experiment. Enrichment was calculated relative to a set of background genes relevant for the experiment. Gene enrichment was also analyzed using Metascape (http://metascape.org)[22].

**ChIP-sequencing**. ChIP experiments were performed as previously described, with the antibodies detailed in the antibody section, and pS213 AR antibody generated previously[18,81]. However, libraries were prepared with the NEBNext Ultra II DNA Library Prep Kit, quantified on Qubit, pooled equimolar, and sequenced on Illumina NovaSeq 6000 SP100 flowcell. Data were analyzed by ROSALIND® (https://rosalind.onramp.bio/), with a HyperScale architecture developed by OnRamp BioInformatics, Inc. (San Diego, CA). Reads were trimmed using cutadapt[67]. Quality scores were assessed using FastQC[68]. Reads were aligned to the Homo sapiens genome build hg19 using bowtie2[82]. Per-sample quality assessment plots were generated with HOMER and Mosaics[83,84]. Peaks were called using MACS26 (with input/IgG controls background subtracted, if provided)[85]. Peak overlaps and differential binding were calculated using the DiffBind R library[86]. Differential binding was calculated at gene promoter sites. Read distribution percentages, identity heatmaps, and FRiP plots were generated as part of the QC step using ChIPQC R library and HOMER[87,88]. Gene enrichment was analyzed using Metascape (http://metascape.org)[22].

**Transwell migration and invasion assays**. Transwell migration and invasion assays were performed using the CytoSelect 24-Well Migration and Invasion Assay (8 um pores) according to manufacturer's instructions. Invasion assays used the membranes coated with the extracellular matrix. Three hundred microliter of cells at 500,000 cells/mL were plated for 48 h for migration assay and 72 h for invasion assay. Images were quantified at 10x magnification by counting cells per field for three separate fields.

**Quantitative RT-PCR and ChIP-PCR**. For qPCR, RNA was extracted using the RNeasy kit (Qiagen) and reverse transcribed using the Verso cDNA Synthesis Kit (ThermoFisher) according to manufacturer instructions. cDNA was amplified containing Fast SYBR Green qPCR Master Mix (ThermoFisher) and qPCR performed using Applied Biosystems Quantstudio 6 Flex Real-Time PCR System. Data was analyzed with the DDCT method normalized to RPL19 with three technical replicates for each experiment. Primers for qPCR were as follows:
RPL19: Fwd CACAAGCTGAAGGCAGACAA, Rev GCGTGCTTCCTTGGTCT TAG
DYNC1I1: Fwd AAAGCTGAGCTAGAGCGCAAA, Rev GTCCTGAACGGGTT CTTTCTTC
MMP7: Fwd GAGTGAGCTACAGTGGGAACA, Rev CTATGACGCGGGAGT TTAACAT
MMP10: Fwd TGCTCTGCCTATCCTCTGAGT, Rev TCACATCCTTTTCGAGG TTGTAG
COL4A6: Fwd CAGCAGCGGGAGAGAAGTC, Rev CAGTAGAGCCAGTGAA TCCT
TLL2: Fwd GCCATGTGGTTGGGTTTTGG, Rev TGTCAAAGTCGTATGTCT CTCCC
PIM1: Fwd GGCTCGGTCTACTCAGGCA, Rev GGAAATCCGGTCCTTCTC CAC
14-3-3 ζ: Fwd CCTGCATGAAGTCTGTAACTGAG, Rev GACCTACGGGCT CCTACAACA
AR: Fwd TACCAGCTCACCAAGCTCCT, Rev GAACTGATGCAGCTCTCTCG

hnRNPK: Fwd CAATGGTGAATTTGGTAAACGCC, Rev GTAGTCTGTACGG AGAGCCTTA
TRIM28: Fwd CGTGTACTGCTGGCCCTATT, Rev AACTCCTGTGGGGAG CTGTA

For ChIP-PCR experiments, ChIP was performed the same way as in the ChIP-sequencing experiments, with corresponding IgGs as negative controls. DNA was amplified containing Fast SYBR Green qPCR Master Mix (ThermoFisher) and primers which were designed based on peaks identified in the ChIP seq data aligned to hg19. Primers are as follows:
MMP7: Fwd CATAGTTTGCCAACTGCTGCT, Rev TCAGTGCTCTTTTCTTAT GAAGATT
TLL2: Fwd TGGGTCATTTCTATGCCAAGCA, Rev TTGCACTGAGCCTTGCA AAC

**Rapid immunoprecipitation and mass spectrometry of endogenous proteins (RIME)**. Immunoprecipitation for RIME was carried out as described for ChIP-Seq. However, based on Mohammed et al, after washing the beads and prior to eluting, beads with antibody-bound proteins were washed twice with 50 mM NH4HCO3 and resuspended in 50 µl 50 mM NH4HCO3 containing 10 ng/µl trypsin/Lys-C (Promega) followed by overnight incubation at 37 °C with vigorous shaking in thermomixer (Eppendorf)[37]. After digestion, beads were pelleted and supernatants containing peptides were transferred to new tubes. After mixing with 5 µl 20% heptafluorobutyric acid, peptides were desalted using C18 spin tips according to manufacturer protocol (Thermo Scientific). Desalted and dried peptides were reconstituted in 12 µl 0.1% formic acid. Peptides concentration was measured spectrophotometrically at 205 nm using NanoDrop One (Thermo Scientific). Peptides were analyzed as described above.

**Statistics and reproducibility**. Statistical testing was completed using Prism 8 for Mac OS X, Version 8.0e. Samples were randomly allocated into experimental groups. Samples were randomly allocated into experimental groups, and no data were excluded in the analysis. Error bars indicate standard deviation. Experiments were repeated 2–3 times and had high reproducibility.

**Reporting Summary**. Further information on research design is available in the Nature Research Reporting Summary linked to this article.

## Data availability

All data generated or analyzed during this study are included in this published article and its supplementary information files (see Supplementary data 2 for primary source data). The mass spectrometry proteomics data have been deposited to the ProteomeXchange Consortium via the PRIDE partner repository with the dataset identifiers PXD023623 and PXD023634 89. The sequencing data have been deposited into Gene Expression Omnibus database (GEO) under reference series GSE181226 (GSE181224 for ChIP-seq data, GSE181225 for RNA-seq data)[89,90]. Uncropped western blots are in Supplementary Figure 5.

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

## Acknowledgements

This work was supported by National Institutes of Health R01CA112226 (S.K.L.), R21CA234291 (M.J.G.), R01GM127267 (E.N.), (T32GM0136573 (S.E.R.), T32CA009161 (S.E.R.). It was also supported by the Blavatnik Family Foundation (E.N.) and Howard Hughes Medical Institute (E.N.). Sequencing data was acquired by NYU Langone's Genome Technology center, which is in part supported by Cancer Center Support Grant P30CA016087 at the Laura and Isaac Perlmutter Cancer Center.

## Author contributions

S.E.R., M.J.G. and S.K.L designed the study. S.E.R. and N.V. carried out experiments and analyzed data. M.J.G., S.K.L. and E.N. provided supervisor oversight. S.E.R. wrote the manuscript. All authors gave input and approved the manuscript.

## Competing interests

The authors declare no competing interests.
