## [Peer Review File · Communications Biology]

Reviewers' comments:

Reviewer #1 (Remarks to the Author):

The androgen receptor is a key therapeutic target in prostate cancer and is therefore of considerable interest to academic, clinical and pharmaceutical research. The receptor protein is also known to be extensively phosphorylated, but there remains a paucity of information on the role of specific phosphorylated residues and the kinases (or phosphatases) responsible for this post-translational modifications. For these reasons the study by Ruff et al is both important and timely. They have studied the interaction between the androgen receptor (AR) and the co-regulatory/ chaperone protein 14-3-3 ζ and phosphorylation by the kinases PIM1, which the authors had previously shown to phosphorylate the AR on serine 213. In the present study they have confirmed this phosphorylation and gone on to demonstrate that PIM1 also phosphorylates 14-3-3 ζ (serine 64) and phosphorylation of both AR and 14-3-3 ζ is important for protein-protein interactions. They have then used RNA-seq (PIM1 overexpression), ChIP (14-3-3 ζ and phosphorylated AR) to interrogate the mechanistic consequences of AR phosphorylation and binding 14-3-3 ζ in response to PIM1 activity. They then go on to select genes involved in extracellular matrix biology to validate the genome-wide studies. The conclusion from these studies is that PIM1 phosphorylation leads to AR and 14-3-3 ζ interactions which regulates the AR transcriptome in prostate cancer cells.

The study is interesting and important as stated earlier, but there are a number of difficulties in the interpretation of the experiments and data presented.

Major Points

1. There is a general lack of information (particularly in figure legends) indicating relative experimental details and importantly the number of experimental replicates throughout the manuscript and lack of statistical analysis (i.e. Figures 4A and B and 5).
2. Figure 1 (Part F) quantitation would be helpful here (and again an indication of replicates) because while the authors conclusion is reasonable, in the absence of PIM1 (lanes 1 and 2) it appears that the phosphorylation mutant receptor actually binds more 14-3-3 ζ ?
3. Figure 2 is RNA-seq data for PIM1 over expression this is then compared with ChIP data for phosphorylated AR and 14-3-3 ζ . The rationale for using 10 % serum rather than charcoal stripped serum with added hormone (Testosterone, DHT or synthetic androgen R1881) is not clear and makes interpretation of the results in Figure 2 and 3 difficult and relating this data to the latter experiments with a selection of genes chosen for validation and biological significance (Figure 5 MMP-10, TLL2). The lack of information throughout the manuscript on the role of hormone is a frustration and needs to be addressed, while it may be assumed that 0.35 nM testosterone (quoted for 10 % serum) activates the AR, it will be important to validate the findings in absence/presence of hormone (Figures 4, 5 and 6).
Minor point- the identification of cellular processes is broader or more general for the ChIP (Figure 3F) than the RNA-seq (Figure 2B) meaning while there is some overlap there also appears to be significant differences.
4. Figure 4 The studies on cell migration and invasion (parts C and D) are interesting and this is related to changes in gene expression (Parts A and B) with over expression and inhibition of PIM1. The direct involvement of the genes studied in migration and invasion, while a reasonable assumption (regulation by PIM1), is not shown experimentally, for example does knock down of CXCL8 and MMP7 or MMP-10 alter the responses in a PIM1-dependent manner?
5. Line 230 'knock down decreased the expression of the genes examined' seems an over interpretation of the data shown (Figure 5A). Unlike Figure 4 DYNC1/1 does not show up regulation by PIM1 overexpression and is not effected by 14-3-3 ζ knock down. Upregulation of TLL2 is more modest and it is not affected by 14-3-3 ζ knock-down.
6. The effects of knock down of AR need to be supported by robust statistical analysis as the changes for DYNC1/1 and CXCL8 appear modest at best (Figure 5B). The effects of using enzalutamide also seem modest or having little effect.
Minor point-labelling of the left hand chart (Figure 5C) vector/PIM1 + 25 μ M enz?
7. Figure 6 the comparison of the two genes, MMP7 and TLL2, upregulated by over expression of PIM1 is interesting as they appear to respond differently to the AR (figure 5) and possible also 14-3-3 ζ (Figure 5). Unfortunately, the analysis is incomplete: in Part B recruitment of 14-3-3 ζ is

confirmed at both genes but no data for AR? Furthermore, in part C we have analysis of MMP7 but not TLL2. From the earlier data it looks like MMP7 maybe activated by AR while TLL2 is repressed, the authors are therefore in a position to shed light on how the receptor function differently at these gene loci. Such an analysis would be more novel and informative than the RIME study, which possibly only picks up the most abundant proteins interacting with AR and 14-3-3 ζ .

8. Minor point- The data in the main manuscript is based primarily on LNCaP cells, there is data with two other prostate cancer cell lines (LAPC4 and 22rv1), this should be incorporate into the main text. There is a growing expectation that multiple cell lines should be used to validate findings. It would also be interesting to include AR negative cell lines to determine receptor-specific response of PIM1 and 14-3-3 ζ .

Reviewer #2 (Remarks to the Author):

Overall this is a well done research paper describing how PIM1 phosphorylation of AR and 14-3-3 lead to AR and 14-3-3 interaction and regulation of gene expression. There are some things that should be addressed:

Western blot quantitation should be in main manuscript, not supplementary data.

The data in Fig 1D is not convincing. This should be redone/replaced or removed.

Statistical analysis of Fig4 and Fig5 is needed.

Supplemental Fig 6 should be in the main manuscript. This is important data for establishing a potential role for TRIM28 and hnRNPK in regulation of AR and 14-3-3.

The n for each experiment should be reported in the figure legend.

Can the authors provide evidence for heterotrimeric complex between PIM1, 14-3-3 and AR? This should be discussed at a minimum.

Point by point response to the review of the manuscript submitted to Communications Biology (COMMSBIO-21-0448-T) entitled “PIM1 phosphorylation of the androgen receptor and 14-3-3 ζ coordinates their interaction and chromatin occupancy to regulate gene transcription in prostate cancer” by Ruff *et al.*

Reviewer 1

The androgen receptor is a key therapeutic target in prostate cancer and is therefore of considerable interest to academic, clinical and pharmaceutical research. The receptor protein is also known to be extensively phosphorylated, but there remains a paucity of information on the role of specific phosphorylated residues and the kinases (or phosphatases) responsible for this post-translational modifications. For these reasons the study by Ruff et al is both important and timely.

Major Points

1. There is a general lack of information (particularly in figure legends) indicating relative experimental details and importantly the number of experimental replicates throughout the manuscript and lack of statistical analysis (i.e. Figures 4A and B and 5).

Thank you for this feedback. We have remedied this by replacing Figures 4 and 5 as well as other applicable panels with figures containing a minimum of 3 biological replicates and thorough statistical analysis. Statistical analysis and replicate numbers are located in the figure legends for each corresponding figure. Additionally, we have removed CXCL8 from our panel of genes examined due to some inconsistencies in the biological replicates.

2. Figure 1 (Part F) quantitation would be helpful here (and again an indication of replicates) because while the authors conclusion is reasonable, in the absence of PIM1 (lanes 1 and 2) it appears that the phosphorylation mutant receptor actually binds more 14-3-3 ζ ?

The quantitation for Figure 1F has been moved to the main figure. The quantified data indicates that, relative to the amount of AR immunoprecipitated in each sample, the phosphorylation mutant does not bind AR more than the WT in the absence of PIM1. See below:

3. Figure 2 is RNA-seq data for PIM1 over expression this is then compared with ChIP data for phosphorylated AR and 14-3-3 ζ . The rationale for using 10 % serum rather than charcoal stripped serum with added hormone (Testosterone, DHT or synthetic androgen R1881) is not clear and makes interpretation of the results in Figure 2 and 3 difficult and relating this data to the latter experiments with a selection of genes chosen for validation and biological significance (Figure 5 MMP-10, TLL2). The lack of information throughout the manuscript on the role of hormone is a frustration and needs to be addressed, while it may be assumed that 0.35 nM testosterone (quoted for 10 % serum) activates the AR, it will be important to validate the findings in absence/presence of hormone (Figures 4, 5 and 6).

We have addressed this point by examining gene expression changes in the absence/presence of hormone. There is new data in Fig. 5C (below), which indicates that the expression of our genes of interest are increased/reduced in a hormone-dependent manner. This indicates that the genes change with the absence/presence of hormone in a complementary manner to AR knockdown (Fig. 5B).

Minor point

4. The identification of cellular processes is broader or more general for the ChIP (Figure 3F) than the RNA-seq (Figure 2B) meaning while there is some overlap there also appears to be significant differences.

We have added the following text to try to explain this discrepancy (lines 171-176):

Although the categories of cellular processes identified in the ChIP-seq analysis appear to be broader than those identified in the RNA-seq analysis, this is likely due to the sheer number of genes showing AR and 14-3-3 ζ occupancy, compared with the relatively few genes with changes in expression at the RNA level. While there was not complete overlap, we chose to focus on genes in these similar classes to determine whether their expression is dependent on 14-3-3 ζ and AR binding.

5. Figure 4 The studies on cell migration and invasion (parts C and D) are interesting and this is related to changes in gene expression (Parts A and B) with over expression and inhibition of PIM1. The direct involvement of the genes studied in migration and invasion, while a reasonable assumption (regulation by PIM1), is not shown experimentally, for example does knock down of CXCL8 and MMP7 or MMP-10 alter the responses in a PIM1-dependent manner?

Thank you for this comment. We agree that this is an interesting point to look into. We decided to perform the migration and invasion studies under 3 conditions: non-targeting siRNA, siRNA

towards genes involved in “migration” (DYNC111 and COL4A6), and siRNA towards genes involved in “invasion” or ECM remodeling (MMP7, MMP10, TLL2). The results we find are below:

For the transwell migration assay (left), knockdown of both sets of genes resulted in reduced migration of the PIM1 over-expressing cells. Although not what we expected, it is possible that all of the genes examined do impact cell migration, whether directly or indirectly.

For the transwell invasion assay (right), we find very few cells invading, even in the non-targeting siRNA control. We have found this previously in our attempts to complete siRNA knockdowns in this cell line for invasion studies. The cells already have very low invasive capacity, and it seems that even the transfection with non-targeting siRNA reduces it further. Therefore, while we see a slight reduction in invasion with the knockdown of pro-invasion genes, we find it hard to make conclusions from this experiment.

Ultimately, we decided not to include these experiments in the manuscript, but we appreciate the recommendation and enjoyed completing this study.

6. Line 230 ‘knock down decreased the expression of the genes examined’ seems an over interpretation of the data shown (Figure 5A). Unlike Figure 4 DYNC1/1 does not show up regulation by PIM1 overexpression and is not effected by 14-3-3 ζ knock down. Upregulation of TLL2 is more modest and it is not affected by 14-3-3 ζ knock-down.

Thank you for this comment – we have clarified the text in multiple places to be more specific regarding which genes are changed in which experiment, as well as adding a minimum of 3 biological replicates with statistical analysis to each experiment.

7. The effects of knock down of AR need to be supported by robust statistical analysis as the changes for DYNC1/1 and CXCL8 appear modest at best (Figure 5B). The effects of using enzalutamide also seem modest or having little effect.

We have added a minimum of 3 biological replicates with statistical analysis to the AR knockdown experiment as well as the enzalutamide experiment. The statistical analysis is fleshed out in the corresponding figure legends.

8. Minor point-labelling of the left hand chart (Figure 5C) vector/PIM1 + 25 μ M enz?

Thanks for pointing this out. We have remedied this labeling error.

9. Figure 6 the comparison of the two genes, MMP7 and TLL2, upregulated by over expression of PIM1 is interesting as they appear to respond differently to the AR (figure 5) and possible also 14-3-3 ζ (Figure 5). Unfortunately, the analysis is incomplete: in Part B recruitment of 14-3-3 ζ is confirmed at both genes but no data for AR? Furthermore, in part C we have analysis of MMP7 but not TLL2. From the earlier data it looks like MMP7 maybe activated by AR while TLL2 is repressed, the authors are therefore in a position to shed light on how the receptor function differently at these gene loci. Such an analysis would be more novel and informative than the RIME study, which possibly only picks up the most abundant proteins interacting with AR and 14-3-3 ζ .

We have completed the analysis requested in Figure 5. First: we validated AR occupancy at MMP7 and TLL2, which is now shown in the revised Fig. 6B (see below):

Next, we examined the 14-3-3 zeta and AR occupancy at TLL2 with SGI1776 and enzalutamide treatment, as well as MMP7 which was previously examined. This is present in the revised Fig. 6C-6D:

We see a similar pattern for TLL2 as with MMP7. This indicates that 14-3-3 ζ binding is likely still AR-dependent at this site. However, changes in the control of the transcription by AR are likely due to changes in the recruitment of co-regulators.

10. *Minor point-* The data in the main manuscript is based primarily on LNCaP cells, there is data with two other prostate cancer cell lines (LAPC4 and 22rv1), this should be incorporate into the main text. There is a growing expectation that multiple cell lines should be used to validate findings. It would also be interesting to include AR negative cell lines to determine receptor-specific response of PIM1 and 14-3-3 ζ.

We have moved this data to the revised Fig. 4C. Additionally, we have completed 2 additional biological replicates for this experiment for a total of 3 biological replicates with corresponding statistical analysis.

Reviewer #2:

Overall this is a well done research paper describing how PIM1 phosphorylation of AR and 14-3-3 lead to AR and 14-3-3 interaction and regulation of gene expression. There are some things that should be addressed:

1. *Western blot quantitation should be in main manuscript, not supplementary data.*

We have moved the Western blot quantitation to the main Figure 1 below each blot.

2. *The data in Fig 1D is not convincing. This should be redone/replaced or removed.*

While we appreciate that this data is not as convincing as the in vitro phosphorylation assay, we believe that there is a clear shift in 14-3-3 zeta in the presence of PIM1. Still, in light of the reviewers comment we moved this data to the supplementary Figure 1.

3. Statistical analysis of Fig4 and Fig5 is needed.

We have completed statistical analysis on Fig 4 and Fig 5. The details of the statistical analysis are in the corresponding figure legends.

4. Supplemental Fig 6 should be in the main manuscript. This is important data for establishing a potential role for TRIM28 and hnRNPK in regulation of AR and 14-3-3.

We have moved this figure to the main manuscript, the revised Figure 8.

5. The n for each experiment should be reported in the figure legend.

The n for each experiment is now present in each figure legend.

6. Can the authors provide evidence for heterotrimeric complex between PIM1, 14-3-3 and AR? This should be discussed at a minimum.

Although we have not observed a heterotrimeric complex between PIM1, AR, and 14-3-3 zeta, we expect that the phosphorylation of AR and 14-3-3 zeta by the PIM1 kinase is a transient interaction. Therefore, we have no reason to believe that all three should be in complex together.

REVIEWERS' COMMENTS:

Reviewer #2 (Remarks to the Author):

In this manuscript Garabedian and co-workers have investigated the interaction between the androgen receptor (AR) and the co-regulatory/ chaperone protein 14-3-3 ζ and the role of phosphorylation by the kinases PIM1. They confirm phosphorylation of the AR on serine 213 and demonstrate that phosphorylation of both AR and 14-3-3 ζ is important for protein-protein interactions. They have then identified and validate target genes for both AR and 14-3-3 ζ , focuses on targets involved in extracellular matrix biology. The conclusion from these studies is that PIM1 phosphorylation leads to AR and 14-3-3 ζ interactions which regulates the AR transcriptome in prostate cancer cells and overall the study is both important and timely.

I appreciate the detailed consideration the author have given to the comments on the original submission and welcome the changes made: including, additional experimental data supporting the conclusions and the additional detailed information and statistical analysis provided in the text and figure legends. I am satisfied the authors have addressed all the concerns raised previously. The changes made strengthen the overall conclusions.

One minor point to note Figure panels 1B and C and transposed in relation to the text (lines 96-97) and figure legend (lines 623-628).

COMMSBIO-21-0448A

Rebuttal

One minor point to note Figure panels 1B and C and transposed in relation to the text (lines 96-97) and figure legend (lines 623-628).

We thank the reviewer for point this out, and have corrected this in the text and figure legend.